Genetic variants of Toll-like receptor 9 are associated with susceptibility to systemic lupus erythematosus in Han Chinese female patients

Zhao Lili 1
Du Shushu 1 2
Xu Wenqi 1
Shi Xiaofei xiaofeis@haust.edu.cn 3
Liu Rongzeng liurz@haust.edu.cn 1
1 Department of Immunology, College of Basic Medicine and Forensic Medicine, Henan University of Science and Technology , Luoyang , China
2 Qingpu Traditional Chinese Medicine Hospital , Shanghai , China
3 Department of Rheumatology and Immunology, The First Affiliated Hospital, and College of Clinical Medicine of Henan University of Science and Technology , Luoyang , China
de Souza Renan
Electronic publication date: 2025 Aug 13
Publication date: 2025
Volume: 13
Electronic Location ID: e19847
Received 2025 Jan 28; Accepted 2025 Jul 15
Copyright: ©2025 Zhao et al.
Copyright year: 2025
Copyright holder: Zhao et al.
License: This is an open access article distributed under the terms of the Creative Commons Attribution License, which permits unrestricted use, distribution, reproduction and adaptation in any medium and for any purpose provided that it is properly attributed. For attribution, the original author(s), title, publication source (PeerJ) and either DOI or URL of the article must be cited.
License URL: https://creativecommons.org/licenses/by/4.0/

Keywords: SLE, TLR9, Single nucleotide polymorphism, Susceptibility

Funding: National Natural Science Foundation of China 81901663 82271851 Young Backbone Teachers Training Program of Henan Province 2023GGJS048 Key Project of Medical Science and Technology Research Program of Henan Province SBGJ202002098 This study received financial support from the National Natural Science Foundation of China (NSFC) through grants numbered 81901663 and 82271851. Additionally, it was supported by Young Backbone Teachers Training Program of Henan Province under grant number 2023GGJS048 and Key Project of Medical Science and Technology Research Program of Henan Province (SBGJ202002098). The funders had no role in study design, data collection and analysis, decision to publish, or preparation of the manuscript.

==============================
Background

Variations in the TLR9 gene have been associated with several autoimmune disorders, but the relationship between TLR9 polymorphisms and systemic lupus erythematosus (SLE) remains controversial. This study aims to evaluate the potential association between three single-nucleotide polymorphisms (SNPs) within the TLR9 gene and susceptibility to SLE in the Han Chinese female population.

Methods

A total of 150 SLE patients and 151 healthy controls of Han Chinese ethnicity were enrolled. Genotyping of TLR9 was performed using sequence-specific primer (SSP) polymerase chain reaction and validated by Sanger sequencing. Associations between the SNPs and SLE susceptibility were analyzed using the chi-square test or Fisher’s exact test. Additionally, correlations between the SNPs and clinical manifestations of SLE were assessed.

Results

The TLR9 rs352139 polymorphism was significantly associated with increased SLE susceptibility in heterozygous (AG vs. AA, OR = 1.79, 95% CI [1.07–2.99], p =  0.025), homozygous (GG vs. AA, OR = 2.11, 95% CI [1.06–4.19], p =  0.033), dominant (GG+AG vs. AA, OR = 1.86, 95% CI [1.15–3.03], p =  0.012), and allele (G vs. A, OR = 1.49, 95% CI [1.07–2.06], p =  0.017) models. Similarly, rs352140 was significantly associated with SLE risk in homozygous (TT vs. CC, OR = 2.47, 95% CI [1.23–4.96], p =  0.010), recessive (TT vs. CC+CT, OR = 2.57, 95% CI [1.35–4.88], p =  0.003), and allele (T vs. C, OR = 1.43, 95% CI [1.03–1.99], p =  0.031) models. Haplotype analysis revealed that haplotype HT1 (C/A/T) had a protective effect against SLE (OR = 0.70, 95% CI [0.506–0.966], p =  0.030), while haplotype HT2 (T/G/T) was positively associated with increased susceptibility (OR = 1.505, 95% CI [1.068–2.121], p =  0.019).

Conclusions

These findings suggest that the TLR9 rs352139 and rs352140 polymorphisms are significantly associated with increased susceptibility to SLE in the Han Chinese population, indicating a potential role of TLR9 in the pathogenesis of SLE.

Introduction

Systemic lupus erythematosus (SLE) is a persistent autoimmune inflammatory condition distinguished by the generation of autoantibodies and widespread inflammation affecting multiple organ systems (Thanou et al., 2021). SLE affects more women than men with a ratio of 9:1, potentially attributed to hormonal influences (Tian et al., 2023). The etiology of SLE remains unclear, although genetic predisposition, environmental factors, and immune dysregulation are believed to play significant roles in its pathogenesis. Among the genetic factors implicated in SLE, Toll-like receptor 9 (TLR9) has been identified as a crucial component due to its essential function in the detection of DNA-containing immune complexes and the subsequent initiation of innate immune responses (Fitzgerald & Kagan, 2020).

TLR9 functions as a pattern recognition receptor that detects unmethylated CpG motifs, which are commonly found in bacterial and viral DNA, as well as in self-DNA under specific pathological conditions (Sha et al., 2021). The recognition of these motifs by TLR9 initiates a series of signaling pathways that culminate in the production of pro-inflammatory cytokines, including interferon-alpha (IFN-α), which play a vital role in the pathogenesis of SLE (Fillatreau, Manfroi & Dorner, 2021; Suspene et al., 2017). Given TLR9’s significant involvement in the activation of the immune system and the development of autoimmunity, genetic variations within the TLR9 gene may have a substantial impact on an individual’s susceptibility to SLE.

The susceptibility of TLR9 gene polymorphisms has been reported in various autoimmune disorders, including type I diabetes, ankylosing spondylitis, and rheumatoid arthritis (Kim et al., 2021; Oliveira-Toré et al., 2019; Wang et al., 2023). Notably, the polymorphisms rs352139, rs352140, and rs5743836 have received significant attention. The rs352139 variant is situated within the first intron of the TLR9 gene and is known to potentially affect TLR9 mRNA levels and protein expression by creating alternative splicing sites (Omar et al., 2012; Redondo et al., 2022). The rs352140 variant, which is a synonymous mutation in the coding region, may influence mRNA stability and the efficiency of translation (Hunt et al., 2009). Meanwhile, rs5743836, located in the promoter region, has the potential to modify transcription factor binding and subsequently alter gene expression levels (Omar et al., 2012). Collectively, these polymorphisms may result in changes to TLR9 function or expression, thereby influencing immune responses and playing a role in the onset and progression of SLE.

Numerous studies have investigated the association between TLR9 gene polymorphisms and SLE across various populations. A case-control study (Hur et al., 2005) indicated that the TLR9 rs352140 polymorphism does not correlate with the onset of SLE, whereas research conducted by Xu et al. (2009) suggested a potential link between the rs352140 polymorphism and increased susceptibility to SLE. These conflicting findings contribute to ongoing debates regarding the relationship between TLR9 polymorphisms and SLE. Furthermore, the influence of TLR9 single-nucleotide polymorphisms (SNPs) on SLE susceptibility may differ among ethnic groups. Therefore, it is crucial to comprehend the functional implications of these SNPs within the framework of TLR9’s role in the immune response to better understand the genetic underpinnings of SLE. The objective of this study was to examine the relationship between TLR9 polymorphisms, specifically rs352139, rs352140, and rs5743836, and the susceptibility of SLE in the Han Chinese female population.

Materials and Methods

Patients and controls

A total of 150 SLE patients were from the Department of Rheumatology and Immunology at the First Affiliated Hospital of Henan University of Science and Technology, including outpatients and inpatients. All patients with SLE meet the diagnostic criteria updated in 1997 by the American College of Rheumatology (Hochberg, 1997). Medical records of individuals diagnosed with SLE were examined. The collected clinical data included anti-dsDNA antibodies, anti-nuclear antibody (ANA), complement C3 and C4, high sensitivity C-reactive protein (hs-CRP), erythrocyte sedimentation rate (ESR), malar rash, arthritis, renal involvement, photosensitivity, oral ulcer, alopecia, leukopenia, nephritis, serositis, and neurologic disorder. This research has received approval from the Ethics Committee at The First Affiliated Hospital of Henan University of Science and Technology (2023-03-K0049). There are 151 cases in the control group, and all individuals have no symptoms or past medical history of rheumatism and immune disorders, additionally, there are no abnormal clinical examination results. The initial study population exhibited a female-to-male ratio of 9:1. Consequently, we opted to include exclusively female participants in both the case and control groups to mitigate potential bias in the results. All individuals involved, both patients and controls, were over the age of 18 and were not related to one another. Written consent was obtained from all participants.

Genomic DNA extraction

Peripheral blood samples collected from participants were treated with EDTA as an anticoagulant and preserved at −80 °C until DNA extraction. Genomic DNA was isolated from one mL of blood samples utilizing the Blood Genomic DNA Extraction Kit (Solarbio, China), following the provided protocol. The concentration of DNA was quantified using a NanoDrop One spectrophotometer (Thermo Fisher Scientific) at a wavelength of 260 nm, while the purity of the DNA was evaluated by calculating the ratio of absorbance at 260 nm to that at 280 nm.

Selection of TLR9 SNPs and genotyping

This research investigated three SNPs within the TLR9 gene, rs352139, rs352140 and rs5743836. The selection of these SNPs was based on a comprehensive search of the dbSNP database for polymorphisms in the human TLR9 gene. Among the identified loci, rs352139 and rs352140 were chosen due to their minimum allele frequency (MAF) exceeding 5% within the Asian population and their frequent examination in relation to various immune diseases, as indicated by a review of existing literature. Furthermore, rs5743836 was included in the study due to its association with multiple immune disorders, as well as its role in the transcriptional regulation and expression of TLR9, which has been shown to enhance TLR9 expression (Carvalho et al., 2011).

The genomic DNA obtained from all participants was subjected to genotyping for three distinct TLR9 SNPs using single specific primer-polymerase chain reaction (SSP-PCR). The primer sequences utilized are detailed in Table 1. PCR amplifications were conducted in a 20 µl reaction mixture comprising 50 ng of template DNA, 10 µl of 2× Taq PCR Master Mix sourced from Sangon Biotech, China, 0.8 µl of gene-specific primers (10 µM each, also from Sangon Biotech, China), and nuclease-free water. The PCR procedure was conducted under the specified parameters: an initial denaturation at 94 °C for 3 min, followed by 30 cycles each of denaturation at 94 °C for 30 s, annealing at 62 °C for 30 s, extending at 72 °C for 40 s and a final extension at 72 °C for 5 min. Subsequently, the amplification products were analyzed using a 2% agarose gel. The presence of specific allele is indicated by the appearance of an amplified PCR product (Fig. S1). Genotyping was conducted on a subset of 10% of the samples in a random manner to verify the consistency of the results, which were found to be entirely consistent. The samples of randomly selected TLR SNPs were sequenced using the Sanger method to confirm the detected genotypes.

Table 1 Primer sequences.

Primers	Sequence	Product size (bp)	
rs352139			
Common	5′-CAAGGAAAGGCTGGTGACAT		
Allele A	5′-AAGTGGAGTGGGTGGAGGTA	270	
Allele G	5′-GTGGAGTGGGTGGAGGTG	268	
rs352140			
Common	5′- TGCGCTACGTGGACCTGTC		
Allele C	5′-TCCAGGGCCTCCAGTCGC	403	
Allele T	5′- TCCAGGGCCTCCAGTCGT	403	
rs5743836			
Common	5′- TCCTCTGCTCAGACCCCTC		
Allele T	5′- ATGAGACTTGGGGGAGTTTT	218	
Allele C	5′-ATGAGACTTGGGGGAGTTTC	218	

Statistical analysis

The genotypes in the control group were assessed for deviation from Hardy-Weinberg equilibrium (HWE) through the application of the chi-square (χ2) test for each SNP, with an HWE analysis with a p value exceeding 0.05, suggesting that the population under study is representative. The χ2 statistic was utilized to compare the frequencies of the allele, genotype and haplotype between the two groups, and the odds ratio (OR) with a 95% confidence interval (CI) was calculated to determine the relative risk. SHEsis Main was used to conduct linkage disequilibrium (LD) analysis and construct the haplotype block pattern (http://analysis.bio-x.cn/SHEsisMain.htm). Pairwise r2 values were calculated on the basis of the genotypes of 151 healthy controls. In order to investigate the association between TLR9 SNPs and particular clinical phenotypes of SLE, the SLE patient cohort was stratified according to various clinical characteristics, including lupus nephritis, cutaneous involvement, and autoantibody profiles. The relationships between TLR9 SNPs and these clinical manifestations were examined utilizing the chi-square test to ascertain genotype-phenotype correlations. Statistical analyses were performed utilizing SPSS software 26.0, with statistical significance determined as a p-value less than 0.05. We employed publicly accessible data from the Genotype-Tissue Expression (GTEx) Expression Quantitative Trait Loci (eQTL) browser for the purpose of conducting eQTL analysis.

Results

Study population characteristics

A total of 301 participants, comprising Han Chinese individuals from northern China, were enrolled in this study. Among these, 150 subjects diagnosed with SLE were exclusively female, with a mean age of 36.5 ± 11.8 years. The healthy control (HC) group consisted of 151 women, with a mean age of 35.4 ± 12.8 years. The clinical characteristics of all participants are detailed in Table 2. Notably, 36.0% of the SLE patients tested positive for anti-dsDNA antibodies, 74.7% were positive for antinuclear antibodies (ANA), 30.0% exhibited decreased levels of complement component C3, 50.7% showed decreased levels of complement component C4, 38.0% had elevated erythrocyte sedimentation rates (ESR), 13.3% presented with increased high-sensitivity C-reactive protein (hs-CRP) levels, and 56.6% had rash, and 51.6% had nephritis.

Table 2 The clinical characteristics of female participants with SLE.

Characteristics	SLE, n (%)	
Rash	69 (56.6%)	
Photosensitivity	28 (23.0%)	
Oral ulcer	12 (9.8%)	
Alopecia	19 (15.6%)	
Leukopenia	34 (27.9%)	
Arthritis	18 (14.8%)	
Nephritis	63 (51.6%)	
Serositis	16 (13.1%)	
Anti-dsDNA Ab	Positive	54 (36.0%)	
Negative	68 (45.3%)	
ANA	Positive	112 (74.7%)	
Negative	10 (6.7%)	
C3	Normal	73 (48.7%)	
Decreased	45 (30.0%)	
C4	Normal	42 (28.0%)	
Decreased	76 (50.7%)	
hs-CRP	Normal	102 (68.0%)	
Increased	20 (13.3%)	
ESR	Normal	65 (43.3%)	
Increased	57 (38.0%)	
Notes.

Abbreviations Anti-dsDNA double-stranded DNA antibody

ANA anti-nuclear antibody

C3 complement 3

C4 complement 4

hs-CRP high sensitivity C-reactive protein

ESR erythrocyte sedimentation rate

Hardy–Weinberg equilibrium

In this study, genotyping was conducted on three SNPs of TLR9 (rs352139, rs352140, rs5743836) in a cohort comprising 150 SLE patients and 151 healthy individuals. The genotype frequencies of the three SNPs were found to adhere to the HWE within the control group (p  > 0.05), indicating that the samples analyzed were considered to be representative. Additional information can be found in Table 3.

Table 3 The distribution of TLR9 allele and genotype frequency and Hardy-Weinberg equilibrium test.

Genotype /Allele	Case (n = 150)	Control (n = 151)	P -value	P /HWE	
rs352139					
AA	40 (26.7%)	61 (40.4%)	0.037	0.833	
AG	81 (54.0%)	69 (45.7%)	
GG	29 (19.3%)	21 (13.9%)	
A	161 (53.7%)	191 (63.2%)	0.017	
G	139 (46.3%)	111 (36.8%)	
rs352140					
CC	54 (36.0%)	61 (40.4%)	0.013	0.352	
CT	61 (40.7%)	74 (49.0%)	
TT	35 (23.3%)	16 (10.6%)	
C	169 (56.3%)	196 (64.9%)	0.031	
T	131 (43.7%)	106 (35.1%)	
rs5743836					
TT	146 (97.3%)	149 (98.7%)	0.541	0.935	
TC	3 (2.0%)	2 (1.3%)	
CC	1 (0.7%)	0	
T	295 (98.3%)	300 (99.3%)	0.250	
C	5 (1.7%)	2 (0.7%)	
Notes.

Bold values represent statistically significant values, p < 0.05.

Abbreviations HWE Hardy-Weinberg equilibrium

Data were expressed as absolute number (n) and percentage (%).

Distribution of TLR9 genotype and allele frequency

Table 3 presented the distribution of genotype and allele frequencies for three SNPs in both SLE individuals and control subjects. Substantial differences were noted in the genotype and allele frequency distribution of TLR9 rs352139 between SLE individuals and the controls (p = 0.037, p = 0.017). Similarly, there were notable differences in genotypic and allelic frequencies of rs352140 between the SLE and the control group (p = 0.013, p = 0.031). However, no statistical difference was noted in the genotype and allele frequency distribution of TLR9 rs5743836 between SLE individuals and the controls.

Association analysis between three SNPs and SLE susceptibility

Five genetic models, including heterozygous, homozygous, dominant, recessive and allele, were applied to assess the association with SLE susceptibility (Table 4). Regarding the rs352139 variant, the AG genotype of rs352139 was found to be associated with an increased risk of SLE in heterozygous model (AG vs. AA, OR = 1.79, 95% CI [1.07−2.99], p = 0.025). The frequency of GG genotype showed significant differences between the patient group and the controls in homozygous (GG vs. AA, OR = 2.11, 95% CI [1.06−4.19], p = 0.033). Under the dominant model, there is a significant difference on the frequency of GG+AG genotype between SLE patients and the controls (GG+AG vs. AA, OR = 1.86, 95% CI [1.15−3.03], p = 0.012). In addition, the presence of G allele was associated with an increased risk of SLE in allelic model (G vs. A, OR = 1.49, 95% CI [1.07−2.06], p = 0.017).

Table 4 Association analysis between three SNPs and SLE susceptibility.

Polymorphisms	Genetic model	Genotype	OR (95% CI)	P -value	
rs352139	Heterozygous	AG vs. AA	1.790 (1.073–2.987)	0.025	
	Homozygous	GG vs. AA	2.106 (1.058–4.194)	0.033	
	Dominant	GG+AG vs. AA	1.864 (1.146–3.032)	0.012	
	Recessive	GG vs. AA+AG	1.484 (0.803–2.741)	0.206	
	Allele	G vs. A	1.486 (1.072–2.058)	0.017	
rs352140	Heterozygous	CT vs. CC	0.931 (0.565–1.534)	0.779	
	Homozygous	TT vs. CC	2.471 (1.232–4.955)	0.010	
	Dominant	TT+CT vs. CC	1.205 (0.756–1.919)	0.432	
	Recessive	TT vs. CC+CT	2.568 (1.352–4.878)	0.003	
	Allele	T vs. C	1.433 (1.032–1.991)	0.031	
rs5743836	Heterozygous	TC vs. TT	1.531 (0.252–9.295)	0.988	
	Homozygous	CC vs. TT	0.993 (0.980–1.007)	0.497	
	Dominant	CC+TC vs. TT	2.041 (0.368–11.315)	0.674	
	Recessive	CC vs. TT+TC	0.993 (0.980–1.006)	0.498	
	Allele	C vs. T	2.542 (0.489–13.207)	0.442	
Notes.

Bold represent statistically significant, p < 0.05.

Abbreviations OR odds ratio

CI confidence interval

As to rs352140, in the homozygote genetic model, TT genotype demonstrates a positive association with the susceptibility to SLE (TT vs. CC, OR = 2.47, 95% CI [1.23−4.96], p = 0.010). In the recessive genetic model, the frequency of TT genotype was found to be greater among the cases compared to the controls, with statistical significance (TT vs. CC+CT, OR = 2.57, 95% CI [1.35−4.88], p = 0.003). In addition, the allele T demonstrated a notable correlation with increased susceptibility to SLE (T vs. C, OR = 1.43, 95% CI [1.03−1.99], p = 0.031). In relation to rs5743836, no statistical differences were detected between the individuals with SLE and the controls in any genetic model.

Analyzing the correlation between haplotypes and SLE susceptibility

Linkage disequilibrium (LD) analysis was carried out for the three SNPs (Fig. 1). LD was present between rs352139 and rs352140 (D′ = 0.706, r2 = 0.464). Association analysis was conducted on SLE patients and healthy controls based on three SNP gene polymorphisms to detect the correlation between each haplotype and SLE susceptibility (Table 5). The frequency of haplotype HT1 (C/A/T) in the controls is significantly higher than that in the cases (OR = 0.7, 95% CI [0.506−0.966], p = 0.030), and this haplotype is negatively correlated with the risk of SLE, which exerting a protective influence against the disease. In contrast to the healthy individuals, the frequency of haplotype HT2 (T/G/T) was found to be significantly higher in the cases (OR = 1.505, 95% CI [1.068−2.121], p = 0.019), leading to an increased susceptibility to SLE.

Figure 1 Haplotype and linkage disequilibrium (LD) coefficients among three SNPs in TLR9 gene.

The value in the LD block represents D’ as a percentage. The redder the color, the stronger the linkage. Four common haplotypes for these polymorphisms were constructed.

Table 5 Haplotype frequencies of TLR9 gene in case and control groups.

Haplotype	rs352139	rs5743836	rs352140	Case (freq)	Control (freq)	P -value	OR (95%CI)	
HT1	A	T	C	141 (47.0%)	170 (56.4%)	0.030	0.700 (0.506–0.966)	
HT2	G	T	T	112 (37.2%)	86 (28.6%)	0.019	1.505 (1.068–2.121)	
HT3	G	T	C	25 (8.3%)	24 (7.9%)	0.826	1.068 (0.594–1.920)	
HT4	A	T	T	17 (5.8%)	20 (6.5%)	0.747	0.896 (0.460–1.745)	
Notes.

Bold values represent statistically significant values, p < 0.05.

Association between investigated SNPs and immune indexes

In order to evaluate the relationship between genotypes and phenotypes in the progression of SLE, we performed a comparative analysis of immune indices among SLE patients with different genotypes of rs352139, rs352140, and rs5743836 (Table 6). A total of 28 patients were excluded from the analysis due to incomplete clinical immune data, resulting in a final sample size of 122 SLE patients. Notably, the analysis revealed no significant differences in various immune indices, including anti-dsDNA antibody levels, ANA, complement components C3 and C4, hs-CRP, and ESR among SLE patients with differing genotypes. Additionally, we examined the association between haplotypes HT1 (C/A/T) and HT2 (T/G/T) and clinical immune indicators in SLE patients. As indicated in Table 7, there was a significant increase in the incidence of alopecia among SLE patients with HT1 (C/A/T) haplotype (p = 0.004). Additionally, serum ANA levels were significantly elevated in SLE patients possessing the HT2 (T/G/T) haplotype (p < 0.001).

Table 6 Association of TLR-9 genotypes to immune indexes of SLE patients.

Characteristics	rs352139 (A > G)	rs352140 (C > T)	rs5743836 (T > C)	
	AA	AG + GG	p value	CC	CT + TT	p value	TT	TC + CC	p value	
	35 (28.6%)	87 (71.4%)		46 (37.7%)	76 (62.3%)		118 (96.7%)	4 (3.3%)		
Anti-dsDNA Ab	13 (37.1%)	41 (47.1%)	0.315	20 (43.5%)	34 (44.7%)	0.892	52 (44.1%)	2 (50.0%)	>0.999	
ANA	31 (88.6%)	81 (93.1%)	0.645	41 (89.1%)	71 (93.4%)	0.619	108 (91.5%)	4 (100.0%)	>0.999	
Decreased C3	13 (37.1%)	32 (36.8%)	0.508	17 (37.0%)	28 (36.8%)	0.950	43 (95.6%)	2 (50.0%)	>0.999	
Decreased C4	20 (57.1%)	56 (64.4%)	0.420	27 (58.7%)	49 (64.5%)	0.432	75 (63.6%)	1 (25.0%)	0.253	
Increased hs-CRP	5 (14.3%)	15 (17.2%)	0.690	7 (15.2%)	13 (17.1%)	0.785	20 (16.9%)	0 (0)	0.831	
Increased ESR	13 (37.1%)	44 (50.6%)	0.179	18 (39.1%)	39 (51.3%)	0.191	54 (45.8%)	3 (75.0%)	0.520	
Rash	22 (62.9%)	47 (54.0%)	0.373	25 (54.3%)	44 (57.9%)	0.702	67 (56.8%)	2 (50.0%)	>0.999	
Photosensitivity	10 (28.6%)	18 (20.7%)	0.349	13 (28.3%)	15 (19.7%)	0.278	28 (23.7%)	0 (0%)	0.573	
Oral ulcer	5 (14.3%)	7 (8.0%)	0.295	4 (8.7%)	8 (10.5%)	>0.999	12 (10.2%)	0 (0%)	>0.999	
Alopecia	10(28.6%)	9(10.3%)	0.012	13 (28.3%)	6 (7.9%)	0.003	19 (16.1%)	0 (0%)	>0.999	
Leukopenia	12 (34.3%)	22 (25.3%)	0.316	14 (30.4%)	20 (26.3%)	0.623	33 (28.0%)	1 (25.0%)	>0.999	
Arthritis	7 (20.0%)	11 (12.6%)	0.300	9 (19.6%)	10 (13.2%)	0.344	18 (15.3%)	1 (25.0%)	0.497	
Nephritis	20 (57.1%)	43 (49.4%)	0.440	23 (50.0%)	40 (52.6%)	0.778	61 (51.7%)	2 (50.0%)	>0.999	
Serositis	6 (17.1%)	10 (11.5%)	0.403	7 (15.2%)	9 (11.8%)	0.592	15 (12.7%)	1 (25.0%)	0.434	
Neurologic disorder	1 (2.9%)	5 (5.7%)	0.672	1 (2.2%)	5 (6.6%)	0.407	6 (5.1%)	0 (0%)	>0.999	
Notes.

Bold values represent statistically significant values, p < 0.05.

Abbreviations Anti-dsDNA double-stranded DNA antibody

ANA anti-nuclear antibody

C3 complement 3

C4 complement 4

hs-CRP high sensitivity C-reactive protein

ESR erythrocyte sedimentation rate

Table 7 Relationship between haplotype and immune indicators in SLE patients.

Characteristics	HT1 (C/A/T)	HT2 (T/G/T)	
	+/+ a	+/−, −/−	p value	+/+ a	+/−, −/−	p value	
	32 (26.2%)	90 (73.8%)		16 (13.1%)	106 (86.9%)		
Anti-dsDNA Ab	12 (37.5%)	42 (46.7%)	0.370	8 (50.0%)	46 (43.4%)	0.147	
ANA	28 (87.5%)	84 (93.3%)	0.511	16 (100.0%)	96 (90.6%)	<0.001	
Decreased C3	12 (37.5%)	33 (36.7%)	0.939	4 (25.0%)	41 (38.7%)	0.245	
Decreased C4	17 (53.1%)	59 (65.6%)	0.195	8 (50.0%)	68 (64.2%)	0.338	
Increased hs-CRP	5 (15.6%)	15 (16.7%)	0.891	4 (25.0%)	16 (15.1%)	0.482	
Increased ESR	11 (34.4%)	46 (51.1%)	0.103	8 (50.0%)	49 (46.2%)	0.184	
Rash	20 (62.5%)	49 (54.4%)	0.430	9 (56.3%)	60 (56.6%)	0.979	
Photosensitivity	10 (31.3%)	19 (21.1%)	0.247	3 (18.8%)	25 (23.6%)	>0.999	
Oral ulcer	4 (12.5%)	8 (8.9%)	0.512	2 (12.5%)	10 (9.4%)	0.657	
Alopecia	10 (31.3%)	9 (10.0%)	0.004	2 (12.5%)	17 (16.0%)	>0.999	
Leukopenia	11 (34.4%)	23 (25.6%)	0.339	5 (31.3%)	29 (27.4%)	0.746	
Arthritis	8 (25.0%)	10 (11.1%)	0.057	1 (6.3%)	17 (16.0%)	0.462	
Nephritis	18 (56.3%)	45 (50.0%)	0.543	9 (56.3%)	54 (50.9%)	0.692	
Serositis	5 (15.6%)	11 (12.2%)	0.624	2 (12.5%)	14 (13.2%)	>0.999	
Neurologic disorder	0 (0)	6 (6.7%)	0.339	2 (12.5%)	4 (3.8%)	0.177	
Notes.

Bold values represent statistically significant values, p < 0.05.

Abbreviations Anti-dsDNA double-stranded DNA antibody

ANA anti-nuclear antibody

C3 complement 3

C4 complement 4

hs-CRP high sensitivity C-reactive protein

ESR erythrocyte sedimentation rate

The impact of rs352139 polymorphism on gene expression levels in different tissues

To elucidate the functional significance of the rs352139 polymorphism, we conducted an analysis of its association with TLR9 expression levels utilizing publicly accessible data from the GTEx eQTL browser. As illustrated in Fig. 2, a notable correlation was identified between TLR9 expression levels and the various genotypes of rs352139 across 751 normal tissue samples. Specifically, the findings indicated that skin tissues harboring the T allele of rs352139 exhibited elevated TLR9 mRNA expression levels (p = 3.57 ×  10−4). The eQTL data were sourced from the GTEx database. Conversely, no significant association was detected between genotype and TLR9 expression levels at the rs352140 locus within the GTEx dataset.

Discussion

SLE is a multifaceted autoimmune condition distinguished by the generation of autoantibodies (Kiriakidou & Ching, 2020). However, research demonstrates the significance of genetic, hormonal, and behavioral influences in the progression of SLE (Lisnevskaia, Murphy & Isenberg, 2014). Recent advancements in genetic studies have led to the identification of several genes linked to the onset and progression of SLE, such as HLA, STAT4, and IRF5 (Hagberg et al., 2018; Li et al., 2020; Molineros et al., 2019). TLRs represent a set of evolutionarily conserved PRRs responsible for sense microbial infection. TLR ligation triggers a cascade of intracellular signals that ultimately lead to the activation of transcription factors such as NF-κB and IRF5 (Yang et al., 2022). Research in murine models of SLE has underscored the critical role of endosomal TLR9 as a primary regulator in the pathogenesis of SLE (Tilstra et al., 2020).

TLR9 activates innate immune responses by recognizing DNAs with unmethylated CpG motifs of prokaryotes and endogenous self-DNA of SLE patients, playing intricate roles in the pathogenesis of SLE (Ohto et al., 2018). Nevertheless, the correlation between TLR9 genetic variations and SLE remains controversial. In the present investigation, we examined the correlation between three SNPs of the TLR9 gene and susceptibility to SLE. Our investigation demonstrates a significant disparity in the frequency distribution of the rs352139 genotype between individuals diagnosed with SLE and the control group, aligning with findings observed in Egyptian populations (Shahin et al., 2016). The presence of the GG genotype is notably associated with an increased susceptibility to SLE, suggesting that individuals with the GG genotype are at a heightened risk for developing the condition. Furthermore, a marked difference in the allele frequency distribution of rs352139 was identified between SLE patients and controls, which contradicts previous studies (Zhang et al., 2014). These discrepancies may be attributed to variations in genotyping methodologies and sample sizes. The results indicate that TLR9 rs352139 may serve as a susceptibility locus for SLE within the Han Chinese female population in northern China, potentially implicating it in the pathogenesis of the disease. Although rs352139 is situated in a non-coding region, the observed variations could influence the binding affinity of transcription factors or other regulatory proteins, thereby modifying gene expression patterns.

The results indicated a notable difference in the distribution of rs352140 allele and genotype frequencies between individuals with SLE and the controls. The T allele of rs352140 was found to significantly increase the susceptibility to SLE. Furthermore, rs352140 was found to be linked with an increased susceptibility to SLE in both recessive and homozygous models. The rs352140 variant is a synonymous mutation located within the exon 2 region of the TLR9 gene. Although it does not modify the amino acid sequence, it has the potential to influence gene function through various mechanisms. At the mRNA level, this variant may affect codon usage or the secondary structure of the mRNA, which in turn can impact translation rates, stability, and splicing processes. At the protein level, it may interfere with translation kinetics, thereby influencing protein folding, ligand binding, and signaling pathways (Im & Choi, 2017). Furthermore, synonymous mutations have been shown to disrupt mRNA structure and diminish translation efficiency, which can subsequently alter disease susceptibility by affecting the expression of mRNA and proteins in candidate genes (Simhadri et al., 2017). Consequently, we propose that the rs352140 synonymous mutation is likely to influence disease susceptibility by modulating TLR9 expression (Giacoletto et al., 2023). Studies have shown that rs352140 is linked to elevated expression of TLR9 in individuals infected with hepatitis virus (Kulmann-Leal et al., 2022). Furthermore, research has demonstrated a significant correlation between rs352140 and lupus nephritis, as well as elevated levels of TLR9 transcripts (Elloumi et al., 2017). The findings suggest that rs352140 may participate in the development of SLE by upregulating TLR9 expression and subsequently modulating the immune reaction. Previous studies have shown differences in the distribution of TLR9 rs352140 gene polymorphism between SLE individuals and the control group within the Han Chinese demographic, which aligns with our study, but opposite with the Zhuang population (Wen et al., 2015). The reason is that different populations exhibit varying genetic backgrounds, which may lead to racial diversity and differences in population stratification.

Figure 2 The relationship between the rs352139 polymorphism and expression of TLR9 examined via eQTL analysis of the GTEx database.

In the skin tissues, the TT genotype was significantly associated with increased expression of the TLR9 gene compared with the AA/AT genotype (p = 3.57 × 10−4).

There is presently no direct experimental evidence to elucidate the impact of rs352139 and rs352140 on TLR9 expression or functionality in SLE. In addition, rs352139 and rs352140 have not yet been identified as genome-wide significant SLE risk loci in large-scale GWAS; these SNPs may exert population-specific effects or reflect modest-risk variants detectable in candidate gene studies; their association with other diseases may suggest broader functional roles in immune and systemic regulation; further replication and functional characterization are warranted to clarify their role in SLE pathogenesis.

rs5743836 is situated within the promoter region of the TLR9, and the variant C allele enhances the gene’s transcriptional activity (Ng et al., 2010). Previously research has indicated a correlation between the C allele of rs5743836 and the onset of RA in female individuals (Gebura et al., 2017). Moreover, there is data suggesting that allele C of rs5743836 increases the susceptibility to spondyloarthritis (SpA) by 1.69-fold regardless of gender or age (Oliveira-Toré et al., 2019). Other researchers have reported a significant correlation between the mutation (CC) genotype or the presence of the TLR9 rs5743836 C allele and the increased susceptibility and severity of coronavirus disease 2019 (COVID-19) infection (Alhabibi et al., 2023). In our investigation, we examined the relationship between the TLR9 polymorphism rs5743836 and the risk of SLE. However, we found no significant association between rs5743836 and SLE susceptibility across any genetic models, a finding that aligns with previous studies (Barbosa et al., 2021). The lack of observed association for rs5743836 may reflect insufficient statistical power rather than a true absence of association with SLE. Consequently, further functional analysis of rs5743836 is warranted to elucidate its precise role in modulating TLR9 expression in the context of SLE.

In order to further investigate the association between genotype and phenotype concerning three SNPs in the context of SLE, we gathered clinical immunological data from 122 SLE patients. This data encompassed various indicators, including anti-dsDNA antibodies, ANA, complement components C3 and C4, hs-CRP, and ESR, which were subsequently analyzed in conjunction with genotyping results. Early complement components play a critical role in the clearance of immune complexes and apoptotic debris, with deficiencies in C4 and C1q frequently linked to the onset of SLE (Macedo & Isaac, 2016). The levels of anti-dsDNA antibodies are significantly correlated with SLE disease activity, exhibiting fluctuations that correspond to the severity of the condition (Mosca et al., 2006). Previous research has indicated a relationship between the presence of anti-dsDNA antibodies and elevated levels of Toll-like receptor 9 (TLR9) in SLE patients (Chauhan et al., 2013). Elevated ESR and reduced CRP levels serve as important markers of inflammation in SLE and are utilized to monitor disease activity. One study has suggested that high hs-CRP levels and increased ESR concentrations are not only closely associated with disease activity but also significantly correlated with damage to major organs or systems (Bertoli et al., 2008). Despite the importance of these immunological indicators in the progression of SLE, our study did not find evidence of a genotype-phenotype impact among SLE patients. Nevertheless, the influence of TLR9 gene mutations on the immune response in SLE patients warrants further consideration. Additionally, due to the absence of clinical data for certain SLE patients in this investigation, it is recommended that the scope of clinical data be expanded for future research.

Genetic variants do not exert great influence by itself and combined analysis can better understand the role of TLR9 variants in SLE. Furthermore, certain haplotypes within the TLR9 gene have the potential to impact the activation of its defense mechanism, thereby influencing the susceptibility or resistance to autoimmune disorders. However, the majority of TLR9 haplotypes remain unidentified at present. Thus, we investigated the haplotype structures of rs352139, rs5743836 and rs352140 variants. The haplotype H1 (C/A/T) is demonstrated to be associated with SLE susceptibility and having a protective effect on SLE. On the contrary, haplotype H4 increases the susceptibility to SLE by 1.5-fold. This suggested that there could be an interaction between these SNPs that cause SLE. Thus, the function of the SLE-predisposing TLR9 haplotype needs to be investigated further. Research on haplotype combinations has revealed that patients with SLE who possess the HT2 (T/G/T) haplotypes exhibit elevated levels of ANA. This suggests that specific mutations in the TLR9 gene may play a role in the manifestation of certain clinical features associated with the disease.

This study examined the possible correlation between SNPs in TLR9 and SLE susceptibility. Like many other genetic association studies, this study possesses certain potential limitations. The current research may be constrained by a limited sample size of participants. This relatively small sample may reduce the statistical power necessary to identify subtle genetic effects, necessitating a cautious interpretation of the findings. Although the p-values for certain SNPs may not retain significance following rigorous correction, they could suggest potential associations that merit further validation in larger, independent cohorts. Additionally, the results may not be applicable to male populations; therefore, future research that includes both genders are essential to examine potential sex-specific associations, particularly in light of the observed gender disparity in the prevalence of SLE and TLR9-related immune responses. Although we incorporated several important clinical parameters, the lack of certain comprehensive clinical measures, such as longitudinal damage indices and cumulative disease activity, restricts a thorough investigation of genotype-phenotype correlations. Future prospective studies that utilize more extensive clinical datasets will enhance our understanding of the clinical significance of the identified genetic associations. Nevertheless, we included participants from a thoroughly characterized demographic and all observable symptoms were validated by seasoned clinicians. The examination of genetic variants linked to TLR9 and their potential impacts on gene functionality and immune regulation could yield significant insights into the mechanisms underlying SLE and aid in the identification of prospective therapeutic targets.

Conclusion

Our data demonstrate that the genetic variants of TLR9, including TLR9 rs352139 and rs352140, are linked to an increased susceptibility to SLE in individuals of northern Chinese Han female descent. TLR9 haplotype HT1 (C/A/T) provides protection from SLE, while haplotype HT2 (T/G/T) is linked to a higher risk of developing SLE. There was a significant increase in the incidence of alopecia among SLE patients with HT1 (C/A/T) haplotype. Meanwhile, the ANA levels were significantly increased in SLE patients carrying HT2 (T/G/T). These findings have the potential to enhance the understanding of the molecular mechanisms involved and to advance the progress of more effective approaches for the diagnosis, prognosis, and management of SLE.

Supplemental Information

Supplemental Information 1 Representative genotyping of TLR9 by PCR-SSP

The band pattern for rs352139, Lanes 1 and 2 indicated GG genotype; Lanes 3 and 4 indicated AG genotype; Lanes 5 and 6 indicated AA genotype; The band pattern for rs352140, Lanes 7 and 8 indicated CC genotype; Lanes 9 and 10 indicated CT genotype; Lanes 11 and 12 indicated TT genotype. The band pattern for rs5743836, Lanes 13 and 14 indicated homozygous CC genotype; Lanes 15 and 16 indicated heterozygous TC genotype; Lanes 17 and 18 indicated homozygous TT genotype. Lane M shows a 600 bp DNA Marker.

Supplemental Information 2 Original image of Fig. S1

Supplemental Information 3 The genotyping results of TLR9 SNPs

The genomic DNA obtained from all participants was subjected to genotyping for three distinct TLR9 SNPs using SSP-PCR. The genotyping results are detailed in Fig. S1.

Additional Information and Declarations

Competing Interests

Author Contributions

Human Ethics

Data Availability

The authors declare there are no competing interests.

Lili Zhao conceived and designed the experiments, performed the experiments, analyzed the data, prepared figures and/or tables, authored or reviewed drafts of the article, and approved the final draft.

Shushu Du performed the experiments, analyzed the data, authored or reviewed drafts of the article, and approved the final draft.

Wenqi Xu performed the experiments, authored or reviewed drafts of the article, and approved the final draft.

Xiaofei Shi conceived and designed the experiments, authored or reviewed drafts of the article, and approved the final draft.

Rongzeng Liu conceived and designed the experiments, authored or reviewed drafts of the article, and approved the final draft.

The following information was supplied relating to ethical approvals (i.e., approving body and any reference numbers):

The research received approval from the Ethics Committee of The First Affiliated Hospital, Henan University of Science and Technology (2023-03-K0049).

The following information was supplied regarding data availability:

Raw data is available in the Supplemental Files.

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
