# Peer review of "Genetic variants of Toll-like receptor 9 are associated with susceptibility to systemic lupus erythematosus in Han Chinese female patients"

_PeerJ, doi:10.7717/peerj.19847_

## Round 0.1 · original submission · Major Revisions

Please address the reviewers’ comments and include a representative agarose gel image in the supplementary material.

·

Basic reporting

The manuscript is generally well-written with professional English and adequate background information. The structure conforms to PeerJ standards. The figures and tables are relevant and well-labeled. However, there are some areas that require improvement:

1. The rationale for selecting these three specific SNPs (rs352139, rs352140, and rs5743836) is not clearly presented in the introduction. The authors should explicitly state why these particular SNPs were chosen over others in the TLR9 gene.

2. Since the study exclusively included female participants (as stated in the Methods section), this should be clearly reflected in the title and throughout the manuscript to avoid overgeneralizing the findings to the entire Chinese population.

Experimental design

The research question is well-defined, and the methodology is generally sound. The genotyping methods and statistical analyses are appropriate. However, there are several limitations:

1. The sample size (150 cases and 151 controls) is relatively small for a genetic association study, which may limit the statistical power to detect modest genetic effects.

2. The clinical characterization of SLE patients could be more comprehensive. Important information about disease onset age, specific clinical manifestations (particularly nephritis or CNS involvement), disease severity measures, and accumulated damage over time are missing.

Validity of the findings

The statistical analyses appear to be appropriate, and the findings are generally well-supported by the data. However, the authors should more clearly acknowledge the limitations of their study, including the female-only cohort and the lack of comprehensive clinical data.

Reviewer 2 ·

Basic reporting

In this manuscript by Zhao et al., the authors investigated the association of 3 TLR9 SNPs (rs352139, rs352140, rs5743836) with SLE risk in a Han Chinese female population. They found that the rs352139 and rs352140 variants were significantly associated with increased SLE risk, while specific haplotypes showed either protective or risk effects. The study is clinically relevant and the authors used robust genotyping methods and comprehensive statistical analysis; however, the work is limited in several ways such as for its small sample size, lack of functional validation, absence of multiple testing correction in the analysis, etc. Please see my detailed comments below.

1. The study uses a relatively small sample size, which may limit statistical power to detect associations, particularly for SNPs with smaller effects. For example, the lack of association observed for rs5743836 with SLE could be due to insufficient power rather than a true absence of effect. Can the authors discuss how the sample size may have influenced their findings, and consider additional analyses (e.g., permutation testing or power calculation) to address this limitation?

2. The authors did not perform any adjustment for multiple comparisons, which could lead to false-positive findings. Could the authors comment on their rationale for not applying a multiple testing correction? How this may affect the interpretation of their results should also be discussed.

3. The authors identified statistical associations of rs352139 and rs352140 with SLE risk, but they didn’t provide any experimental evidence or literature citation to demonstrate how these variants might affect TLR9 expression, splicing, immune signaling, etc., in the context of SLE. This limits the biological interpretation of the findings, which makes it hard to determine whether these are the actual causal variants or SNPs that are simply linked to causal variants. Also, the authors mentioned that rs352140 is a synonymous variant but do not refer to any published work or any experimental evidence to show that it alters mRNA stability or translation efficiency.

4. Have these SNPs, or any other linked SNPs, been reported in GWAS as SLE risk loci? The authors should check GWAS data to rule out other strongly associated TLR9 SNPs present nearby. Also, if these SNPs have been associated with other diseases (for example, rs352140 has been linked with anxiety and body height; rs352139 with bipolar disorder and gastroesophageal reflux disease), can the authors provide any biological interpretation that might explain their involvement in SLE?

5. Do rs352140 or the other variants act as eQTLs influencing TLR9 expression in relevant tissues? I suggest that the authors check the GTEx database to see whether these SNPs are associated with an increase or decrease in TLR9 mRNA, as this could provide important insight about their relevance in SLE pathogenesis.

6. Alongside overall SLE risk, I think it would be helpful if the authors analyze whether these SNPs are associated with specific clinical features of SLE, such as anti-dsDNA, anti-Sm, elevated IFN-α, etc. This would add more biological insight to the study.

Experimental design

Please see my comments in the basic reporting section.

Validity of the findings

Please see my comments in the basic reporting section.

Additional comments

Please see my comments in the basic reporting section.

---

## Round 0.2 · accepted · Accept

Authors have addressed all of the reviewers' comments.

Reviewer 2 ·

Basic reporting

I am satisfied with the revised manuscript and have no further comments. The updated version is stronger and more comprehensive, effectively addressing the statistical concerns and incorporating findings from GTEx. I recommend the manuscript for acceptance. Congratulations.

Experimental design

Please refer to my comments above.

Validity of the findings

Please refer to my comments above.

Additional comments

Please refer to my comments above.